# Drug Development in Inflammatory Bowel Diseases: What Is Next?

**DOI:** 10.3390/ph18020190

**Published:** 2025-01-30

**Authors:** Lorenzo Petronio, Arianna Dal Buono, Roberto Gabbiadini, Giulia Migliorisi, Giuseppe Privitera, Matteo Ferraris, Laura Loy, Cristina Bezzio, Alessandro Armuzzi

**Affiliations:** 1IBD Center, Department of Gastroenterology, IRCCS Humanitas Research Hospital, via Manzoni 56, Rozzano, 20089 Milan, Italy; lorenzo.petronio@humanitas.it (L.P.); arianna.dalbuono@humanitas.it (A.D.B.); roberto.gabbiadini@humanitas.it (R.G.); giulia.migliorisi@humanitas.it (G.M.); giuseppe.privitera@humanitas.it (G.P.); matteo.ferraris@humanitas.it (M.F.); laura.loy@humanitas.it (L.L.); cristina.bezzio@hunimed.eu (C.B.); 2Department of Biomedical Sciences, Humanitas University, via Rita Levi Montalcini 4, Pieve Emanuele, 20072 Milan, Italy

**Keywords:** Crohn’s disease, ulcerative colitis, anti-TL1A antibodies, obefazimod, S1P modulators, anti IL-23, NX-13

## Abstract

**Background/Objectives**: Inflammatory bowel diseases (IBDs), which include Crohn’s disease (CD) and ulcerative colitis (UC), are chronic conditions requiring long-term therapy to maintain remission and improve quality of life. Despite the approval of numerous drugs, IBD continues to present treatment challenges. This review aims to summarize novel therapeutic target agents in phases II and III of development, including sphingosine-1-phosphate receptor modulators (S1P), anti-interleukin-23 (IL-23), and other small molecules and monoclonal antibodies currently under investigation (e.g., anti-TL1A, obefazimod, NX-13, RIPK-inhibitors). **Methods**: A comprehensive literature search was conducted up to December 2024 to identify relevant articles published in English over the past three–five years, focusing on phase II/III studies for UC and CD. The search included databases such as PubMed, Google Scholar, and the ClinicalTrials.gov portal. **Results**: Clinical trials underline the potential of novel immunomodulators, including anti-TL1A, obefazimod, NX-13, RIPK inhibitors, and anti-IL-23p19 agents, as promising therapeutic options for IBD. Anti-IL23p19 therapies, such as risankizumab and mirikizumab, alongside guselkumab, exemplify this class’s growing clinical relevance. While some are already in clinical use, others are nearing approval. **Conclusions**: Ongoing research into long-term safety and the development of personalized treatment strategies remains pivotal to enhance outcomes. Patient stratification and the strategic positioning of these therapies within the expanding treatment landscape are critical for optimizing their clinical impact.

## 1. Introduction

Inflammatory bowel diseases (IBDs), including Crohn’s disease (CD) and ulcerative colitis (UC), are idiopathic, immune-mediated conditions affecting the digestive system. They are characterized by a chronic remitting–relapsing course, with phases of variable duration of quiescence interrupted by episodes of exacerbation [1,2,3]. The etiology of these diseases is multifactorial, resulting from a complex interaction between genetic factors, environmental factors, and immune factors, through the production of pro-inflammatory cytokines that are released locally in the intestine by cells of the innate and adaptive immune systems [4,5]. The prevalence rates of IBD have shown relative stability in industrialized regions, such as North America and Western Europe. In contrast, they have steadily increased in rapidly industrializing countries, notably China [6]. Currently, the treatment of moderate-to-severe IBD encompasses conventional and advanced targeted therapies [7]. Approved advanced targeted therapies include tumor necrosis factor alpha inhibitors (TNFi), anti-integrin α4β7 [8,9], anti-interleukin (IL) 12/23 [10,11,12], and Janus kinase (JAK) inhibitors [13,14,15]. Despite the availability of multiple therapies, a considerable proportion of patients remain unresponsive to treatment. Clinical and real-life data indicate that the efficacy of any single drug rarely exceeds 60% of treated patients [16]. Indeed, approximately 30% of patients can exhibit primary failure to TNFi, and one-third of patients of those who initially respond can relapse in the long term, despite increased dosing [17]. This critical limitation has driven the scientific community to deepen its investigation into the mechanisms of intestinal inflammation underlying IBD to expand the therapeutic arsenal available to physicians and patients [18,19]. This review aims to explore new and emerging therapies for IBD, focusing on the most recent phase II–III studies of novel drugs. These include interleukin (IL) 12/23 inhibitors, TL1A inhibitors, selective up-regulators of miR-124 (e.g., obefazimod), activators of NLRX1 (e.g., NX-13), RIPK inhibitors, and sphingosine-1-phosphate (S1P) receptor modulators (immunomodulators).

## 2. Methods

A comprehensive search of PubMed/MEDLINE, Google Scholar, and the ClinicalTrials.gov portal was conducted up to December 2024 to identify studies exploring novel and emerging therapies for IBD. This included treatments currently in late-phase II and phase III trials that are expected to become globally available within the next few years. The search strategy included the following text words and corresponding Medical Subject Heading/Entree terms: “inflammatory bowel disease” OR “IBD” OR “ulcerative colitis” OR “Crohn’s disease” AND “Janus kinase (JAK) inhibitors”, OR “TL1A”, OR “sphingosine-1 phosphate (S1P) receptor modulators”, OR “Micro-RNA-124 (miR-124) up-regulators”, OR “anti-IL23”. Additional relevant publications were identified through a manual review of abstracts from the annual meetings of Digestive Disease Week, the American College of Gastroenterology, the European Crohn’s and Colitis Organization, and United European Gastroenterology Week. No restrictions were applied to publication date.

## 3. Crohn’s Disease

### 3.1. Anti-Interleukin-23 (IL-23)

For years, research has focused on interleukin-23 (IL-23), a member of the IL-12 cytokine family, because of its critical role in the pathogenesis of IBD. IL-23 promotes the survival and differentiation of T-helper 17 (Th17) cells, which upon activation produce pro-inflammatory cytokines such as IL-17 and IL-22. Aberrant activation of the IL-23 pathway has been implicated in driving the chronic inflammation in IBD. Exploiting the role of IL-23 in driving chronic inflammation, drugs targeting this pathway have been developed to mitigate inflammation and improve disease outcomes. The first drug explored in this context was ustekinumab, a monoclonal antibody that neutralizes both IL-12 and IL-23 by binding to their shared p40 subunit, thereby inhibiting their pro-inflammatory activity [20,21].

*Risankizumab* is a humanized monoclonal antibody specifically designed to target the p19 subunit of IL-23. By blocking this subunit, risankizumab prevents the activation of downstream cytokine signaling, effectively disrupting the inflammatory cascade driven by IL-23. The phase III ADVANCE trial (NCT03105128) was a multicenter, randomized, double-blind, placebo-controlled induction study evaluating the efficacy and safety of risankizumab as an induction therapy in patients with moderately to severely active CD who were unresponsive to conventional therapies. A total of 931 patients were randomized to receive risankizumab at 600 mg (*n* = 373), risankizumab at 1200 mg (*n* = 372), or a placebo (*n* = 186) for 12 weeks. Clinical remission rates, based on the Crohn’s Disease Activity Index (CDAI), were 45% in the 600 mg group, 42% in the 1200 mg group, and 25% in the placebo group. Endoscopic response rates similarly favored risankizumab over the placebo [22].

The phase III MOTIVATE trial (NCT03104413) was a multicenter, randomized, double-blind, placebo-controlled induction study evaluating the efficacy and safety of risankizumab as an induction therapy in patients with moderately to severely active CD who had failed prior biologic therapies. A total of 618 patients were randomized to receive risankizumab at 600 mg (*n* = 206), risankizumab at 1200 mg (*n* = 205), or a placebo (*n* = 207) for a total of 12 weeks. At week 12, clinical remission rates based on the CDAI were 42% for the 600 mg dose, 40% for the 1200 mg dose, and 20% for the placebo group. Endoscopic response rates also confirmed the efficacy of risankizumab over the placebo. In both ADVANCE and MOTIVATE, the most common adverse events were headache and nasopharyngitis [22].

The phase III FORTIFY trial (NCT03105102) was a multicenter, randomized, double-blind, placebo-controlled maintenance and open-label extension study to assess the efficacy and safety of risankizumab as a maintenance therapy in patients with moderately to severely active CD who achieved clinical remission in the ADVANCE and MOTIVATE studies. Participants were randomized to receive risankizumab at 180 mg or 360 mg for 52 weeks. Clinical remission rates at week 52 were 52% in the 360 mg group and 55% in the 180 mg group, with endoscopic remission rates also favoring risankizumab over the placebo. Common adverse events included CD exacerbations, arthralgia, and headache. It should be noted that the study included only patients who had responded to the induction period with risankizumab, thereby excluding non-responders. This could potentially overestimate the treatment’s efficacy in the general population [23].

The phase IIIb SEQUENCE trial (NCT04524611) was a multicenter, randomized, blinded study designed to compare the efficacy of risankizumab versus ustekinumab in adult patients with moderately to severely active CD who had failed anti-TNF therapy. Participants were randomized to receive risankizumab (*n* = 255) or ustekinumab (*n* = 265) for 48 weeks. Primary endpoints included clinical remission (defined as CDAI < 150) at week 24 and endoscopic remission at week 48 (defined as a score ≤4, a decrease of ≥2 points from baseline, and no subscore >1 in any individual variable of the Simple Endoscopic Score for Crohn’s Disease [SES-CD]). At the end of the study, risankizumab demonstrated equivalence to ustekinumab in achieving clinical remission at week 24 and superiority in achieving endoscopic remission at week 48. Among the limitations of this study, it is important to consider that 27.2% of the participants had discontinued ustekinumab compared to 9.8% who had discontinued risankizumab. This may have influenced the interpretation of the results regarding efficacy [24]. In addition, patients taking corticosteroids at baseline had higher rates of clinical remission with risankizumab compared to ustekinumab at week 48 (56.9% vs. 31.0%) [25].

*Mirikizumab* is a humanized IgG4 monoclonal antibody directed against IL-23p19. The phase II SERENITY trial (NCT02891226) was a multicenter, randomized, parallel-arm, placebo-controlled study designed to evaluate the efficacy and safety of mirikizumab in patients with active CD. A total of 191 patients were enrolled and randomized to receive a placebo (*n* = 64), or 200 mg (*n* = 31), 600 mg (*n* = 32), or 1000 mg (*n* = 64) of mirikizumab intravenously during a 12-week induction period. At the end of this period, endoscopic response (defined as a reduction of at least 1 point in SES-CD) was observed in 10.9% of the placebo group, 25.8% of the 200 mg group, 37.5% of the 600 mg group, and 43.8% of the 1000 mg group. Mirikizumab responders were then randomized to receive 300 mg of the drug either intravenously or subcutaneously during a 40-week maintenance phase. At week 52, an endoscopic response was achieved in 58.5% of patients receiving intravenous (IV) mirikizumab and 58.7% of patients receiving the subcutaneous (SC) formulation. The most common adverse events reported were headache, weight gain, and nasopharyngitis [26].

The phase III VIVID I trial (NCT03926130) was a multicenter, randomized, double-blind, placebo- and active-controlled, treat-through study assessing the efficacy and safety of mirikizumab in patients with moderately to severely active CD who had inadequate response to prior biological therapy. A total of 1065 patients were randomized to receive mirikizumab (*n* = 579), ustekinumab (*n* = 287), or a placebo (*n* = 199) for a total treatment period of 52 weeks (12-week induction period followed by 40-week maintenance period). The primary endpoints of the study were clinical response (defined as ≥30% reduction in stool frequency or abdominal pain score) at week 12, clinical remission (defined as CDAI < 150) at week 52, and endoscopic response (defined as ≥50% reduction in SES-CD score from baseline) at week 52. At week 52, 45.4% of patients treated with mirikizumab achieved clinical remission compared to 19.6% in the placebo group. Endoscopic response was also superior in the mirikizumab group (38%) compared to the placebo (9%). Adverse events were less common in the mirikizumab group than in the placebo group. The study compared the efficacy and safety of mirikizumab with that of ustekinumab. Mirikizumab was shown to be non-inferior to ustekinumab in inducing and maintaining clinical remission and endoscopic response at week 52. Both drugs showed a similar safety profile, with minimal differences in adverse events [27].

The VIVID II study (NCT04232553) is an ongoing phase III, multicenter, open-label, long-term extension study designed to evaluate the continued efficacy and safety of mirikizumab. Results are expected in 2026 [28].

Guselkumab is a human monoclonal antibody that selectively inhibits the p19 subunit of IL-23. The GALAXI I trial (NCT03466411) is a multicenter, randomized, double-blind, placebo- and active-controlled, parallel-group program consisting of GALAXI 1, a phase II study, and GALAXI 2 and 3, two phase III studies. This program evaluated the effects of guselkumab in patients with CD who had inadequate response to prior biologic therapy. In GALAXI 1, patients were randomized to receive guselkumab IV at doses of 200 mg, 600 mg, or 1200 mg, or a placebo. After the 12-week induction period, a significant reduction in the CDAI score was observed in each guselkumab group compared to the placebo group. Guselkumab demonstrated a good safety profile, with headache and nasopharyngitis being the most reported adverse events [29].

After 12 weeks of induction therapy, two subcutaneous maintenance doses were evaluated up to week 48: a dose of 100 mg every 8 weeks (following the 200 mg IV induction dose) and a dose of 200 mg every 4 weeks (following the 600 mg and 1200 mg IV induction doses). At the end of the study period, clinical remission (defined as CDAI <150) was achieved by 64% of patients in the 200→100 mg group, 73% in the 600→200 mg group, and 57% in the 1200→200 mg group. Meanwhile, endoscopic response (50% improvement in SES-CD from baseline) and endoscopic remission (SES-CD ≤ 2) were achieved in 44% and 18% (200→100 mg), 46% and 17% (600→200 mg), and 44% and 33% (1200→200 mg), respectively. Analysis of the results showed that guselkumab demonstrated durable clinical and endoscopic efficacy up to 48 weeks with a favorable safety profile [30].

In GALAXI 2 and 3, recently highlighted at Digestive Disease Week 2024, patients were randomized to receive ustekinumab or guselkumab at different dosing regimens, specifically 100 mg of guselkumab SC every 8 weeks (following a 200 mg IV induction dose), 200 mg SC of guselkumab every 4 weeks (following a 200 mg IV induction dose), and 90 mg SC of ustekinumab every 8 weeks (following a 6 mg/kg IV induction dose). At week 48, clinical remission (defined as CDAI < 150) was achieved by a higher proportion of patients who received guselkumab (65.4% in the 100 mg SC group and 70.3% in the 200 mg SC group) compared to those who received ustekinumab (62.9%). Similar results were observed for endoscopic remission (defined as SES-CD ≤ 2), where guselkumab showed superiority over ustekinumab [31].

The GRAVITI study (NCT05197049) is a multicenter, randomized, double-blind, placebo-controlled study designed to evaluate the efficacy and safety of guselkumab SC administration in patients with moderate-to-severe active CD. A total of 347 patients were randomized 1:1:1 to receive guselkumab at 400 mg SC at weeks 0, 4, and 8, followed by 200 mg SC every 4 weeks (*n* = 115), guselkumab at 400 mg SC at weeks 0, 4, and 8, followed by 100 mg SC every 8 weeks (*n* = 115), and a placebo (*n* = 117). At the end of the study period, clinical remission (defined as CDAI < 150) and endoscopic response (defined as a ≥50% reduction in SES-CD score from baseline) were achieved in 56.1% and 41.3% of the guselkumab groups, respectively, compared to 21% of the placebo group [32].

Ongoing trials include the FUZION CD trial (NCT05347095), a phase III study focusing on the efficacy of guselkumab efficacy in patients with fistulizing and perianal disease [33]. Table 1 highlights the ongoing trials investigating novel therapies in CD.

### 3.2. Anti-TL1A Antibodies

TL1A (tumor necrosis factor-like cytokine 1A) is a cytokine, typically expressed on the surface of antigen-presenting cells (APCs), such as dendritic cells, and belongs to the tumor necrosis factor (TNF) ligand superfamily. Its overexpression leads to the binding of TL1A to death receptor 3 (DR3), which is expressed on the surface of lymphocytes and other immune cells. The TL1A–DR3 interaction stimulates cell proliferation and the production of multiple cytokines by T cells, thereby enhancing the Th1, Th2, and Th17 immune responses [34].

In murine models of IBD, TL1A involvement was associated with upregulation of the TL1A–DR3 complex in affected cells, whereas a decrease in its expression was observed following effective anti-inflammatory treatments [35]. Based on these studies, the use of anti-TL1A monoclonal antibodies has been shown to result in a significant reduction in symptoms, highlighting the therapeutic potential of these strategies for the treatment of IBD [36]. In addition, TL1A plays a critical role in stimulating intestinal myofibroblasts to produce collagen through the TGF-β1/Smad3 signaling pathway, contributing to the development of intestinal fibrosis. This represents a common and challenging complication in IBD patients [37,38].

Tulisokibart is a monoclonal antibody targeting the TL1A–DR3 complex, currently being investigated in CD. The APOLLO-CD study was a multicenter, open-label phase IIa trial designed to evaluate the efficacy and safety of tulisokibart as a 12-week induction treatment for moderately to severely active CD. A total of 55 patients were enrolled and received staggered doses of tulisokibart: 1000 mg on day 1, followed by 500 mg at weeks 2, 6, and 10. By the end of the induction period, 26% of patients achieved an endoscopic response (defined as a ≥50% improvement in the SES-CD score), while 49.1% achieved clinical remission (defined as a CDAI score of <150 points). No serious treatment-related adverse events were reported [39,40]. Following the 12-week treatment, tulisokibart responders (defined as those showing a ≥100-point reduction in CDAI from baseline or achieving a CDAI <150 at week 12) were enrolled in the APOLLO-CD extension study. Non-responders were withdrawn from the study. Responders (*n* = 37) were randomized to receive tulisokibart at either 100 mg (*n* = 19) or 250 mg (*n* = 18) up to week 170. Although the study is ongoing, preliminary efficacy results up to week 50 have been published. At week 50, both dose groups demonstrated sustained clinical and endoscopic benefits, with the higher-dose group showing superior results. Patients receiving 250 mg had a greater reduction in C-reactive protein (CRP) levels compared to the 100 mg group. Mild to moderate adverse events were reported in 83% of participants in the 250 mg group and 84% in the 100 mg group, while serious adverse events were observed in 6% (one participant) and 11% (two participants), respectively [41]. Although these early results are promising, larger studies are needed to confirm the findings [41].

The ongoing NCT06430801 study is a phase III, double-blind, placebo-controlled trial designed to further evaluate the efficacy and safety of tulisokibart in patients with moderately to severely active CD. Approximately 1200 patients will be enrolled, and the trial will assess the proportion of patients achieving clinical remission and endoscopic response at week 52. The study started in June 2024, and final results are expected to take several years to become available [42]. 

The RELIEVE UCCD study (NCT05499130) is an ongoing phase IIb, randomized, double-blind, dose-ranging study designed to evaluate the efficacy, safety, and tolerability of duvakitug in patients with moderately to severely active CD. Participants were randomized to receive low and high doses of duvakitug and a placebo in a 1:1:1 ratio for 14 weeks. At the end of the study period, the primary endpoints were achieved in 26.1%, 47.8%, and 13% of patients in the study groups, respectively. Duvakitug was generally well tolerated, with adverse event rates similar to the placebo [43].

The NCT05910528 study is an ongoing phase II, randomized, double-blind, multicenter, induction and maintenance study designed to evaluate the safety and efficacy of RO7790121 in patients with moderately to severely active CD. Preliminary results will be available in 2030 [44].

### 3.3. Obefazimod

Obefazimod (ABX464) is an anti-inflammatory quinoline currently under investigation for IBD. Notably, this drug is not new in the scientific literature, having been initially developed as a treatment for HIV with promising results [45]. Its mechanism of action involves immune modulation through its interaction with microRNA-124 (miR-124), a molecule recognized for its central role in the inflammatory cascade. MicroRNAs (miRs) are small, non-coding RNA oligonucleotides that regulate gene expression and have been implicated in the pathogenesis of several inflammatory diseases. Numerous miRs are deregulated in IBD, including miR-124. Researchers suggest that the downregulation of miR-124 significantly increases STAT3 protein expression, thereby increasing the transcription of its downstream target genes [45,46]. The mechanism of action of ABX464 involves binding to the cap-binding complex (CBC), a protein complex at the 5′ end of RNA, to selectively enhance the splicing of a long non-coding RNA. This generates miR-124, which downregulates several pro-inflammatory cytokines (including TNFα, IL-6, and IL-17), reduces Th17+ cells, and contributes to improving epithelial barrier function, thereby reducing intestinal permeability [46,47]. Preclinical studies have highlighted obefazimod’s remarkable ability to alleviate intestinal inflammation in murine models of dextran sulfate sodium (DSS)-induced colitis. This is achieved by significantly reducing pro-inflammatory cytokines (like TNF-α, IL-6, and MCP-1) and boosting IL-22 expression, a cytokine crucial for tissue repair. These findings have paved the way for future clinical studies to evaluate its safety and efficacy in patients with IBD [48]. The phase IIb clinical trial ENHANCE-CD (NCT06456593) is a double-blind, placebo-controlled study evaluating the efficacy, safety, pharmacokinetics, and pharmacodynamics of three doses of obefazimod in moderately to severely active CD. The study design consists of a 12-week induction phase followed by a 40-week maintenance phase and a 48-week extension phase [49]. Recruitment started in September 2024, with primary completion targeted by December 2026 and full-study completion expected by April 2028.

The ongoing phase IIb/III trial NCT03905109, a multicenter, randomized, placebo-controlled study, is designed to evaluate the efficacy and safety of ABX464 in patients with moderately to severely active CD. This trial includes an induction phase followed by a maintenance phase, with a total treatment duration of 52 weeks [50].

### 3.4. S1P Receptor Modulators

S1P is a bioactive lipid metabolite produced through the phosphorylation of sphingosine, a process catalyzed by two distinct enzymes: sphingosine kinase 1 (SphK1) and sphingosine kinase 2 (SphK2). S1P exerts its biological functions by binding to G-protein-coupled receptors (S1P1–S1P5), which are differentially expressed in different tissues. Among these receptors, S1P1 is the most ubiquitously expressed and is found on the surface of immune and endothelial cells. The interaction between S1P and S1P1 (S1P–S1PR1) plays a central role in inflammation. This pathway promotes lymphocyte migration from lymphoid organs (e.g., spleen and lymph nodes) to peripheral tissues, enhances Th1 and Th17 differentiation, inhibits regulatory T cell (Treg) activity, and disrupts the intestinal barrier integrity [20,46,51]. The S1P/S1PR axis is an attractive therapeutic target for the treatment of several immune-mediated diseases, including IBD. S1P receptor (S1PRs) agonists have been developed to modulate this pathway. These agonists bind to S1PRs and induce their internalization, ubiquitination and subsequent degradation. This process effectively blocks lymphocyte migration and attenuates inflammatory responses, providing a novel mechanism for controlling inflammation in IBD [52].

Amiselimod is a drug that modulates the S1P1 receptor. A phase IIa multicenter, randomized, double-blind, placebo-controlled, parallel-group study was conducted to evaluate the safety, tolerability, and clinical efficacy of amiselimod administered once daily for 12 weeks in patients with moderate-to-severe CD. In this trial, 40 patients received 0.4 mg of amiselimod and 38 patients received a placebo. At the end of the 12-week treatment period, amiselimod did not demonstrate superiority over the placebo in inducing a clinical response, defined as a reduction in the CDAI. Similarly, no significant differences were observed in serum CRP or fecal calprotectin levels. However, amiselimod was well tolerated and effectively reduced peripheral lymphocyte counts, reflecting its pharmacodynamic activity. Among the limitations of the study, it can be noted that patients with ileal lesions exhibited lower plasma concentrations of amiselimod compared to those with colonic lesions. This suggests that drug absorption may have been impaired by inflammation in the small intestine. Additionally, the clinical response rate to the placebo (54.1%) was higher compared to other similar studies, which may have masked the potential effects of amiselimod [53]. Moreover, amiselimod was designed to have a more favorable safety profile by reducing first-dose heart rate effects, but this slower pharmacokinetic profile may have contributed to a delayed onset of therapeutic action, making it less competitive.

Ozanimod is a selective oral immunomodulator targeting S1P1 and S1P5 receptors. Ozanimod is not yet FDA-approved for CD, but promising studies have been conducted. The STEPSTONE study, an uncontrolled, multicenter phase II trial, enrolled 69 patients with moderate-to-severe CD. Participants initiated treatment with 0.25 mg of ozanimod daily for 4 days, followed by 0.5 mg for 3 days, and then 1 mg daily for 11 weeks. At week 12, endoscopic response, clinical remission, and clinical response were observed in 23.2%, 39.2%, and 56.5% of patients, respectively [54]. The NCT05470985 study is a phase II/III, multicenter, randomized, double-blind trial designed to evaluate the efficacy, safety, pharmacokinetics, and pharmacodynamics of oral ozanimod (RPC1063) in pediatric patients with moderate-to-severe CD who have failed to respond adequately to conventional therapies. This study will provide valuable insights into the effects of ozanimod in younger populations and is expected to be completed by 2032 [55]. The phase III YELLOWSTONE project consists of two randomized, double-blind, placebo-controlled induction studies (NCT03440372 and NCT03440385), a maintenance study (NCT03464097), and an open-label extension study (NCT03467958). Primary endpoints include endoscopic response and clinical remission, assessed using SES-CD and CDAI, respectively. These studies are ongoing, and results are expected around 2026. The goal of this project is to demonstrate sustained efficacy and acceptable safety to support FDA approval of ozanimod for the treatment of CD [56].

Etrasimod is an oral selective agonist of the S1P1, S1P4, and S1P5 receptors, offering a novel approach to reducing inflammation in IBD. The CULTIVATE project is a phase II/III program assessing the efficacy, safety, and tolerability of etrasimod in adult patients with moderate-to-severe CD who are refractory to existing therapies. The program comprises five substudies focusing on induction dosing (substudies A and 1), maintenance (substudy 3), therapeutic response (substudy 2), and long-term effects (substudy 4). Preliminary results from substudy A showed clinical remission rates of 31% and 44% with 2 mg and 3 mg doses of etrasimod, respectively, highlighting its potential in the treatment of CD (Table 1). Results from other substudies are pending [57,58].

### 3.5. Combination Therapies

Combination therapy is one of the emerging strategies for treating CD, aiming to enhance inflammation control, reduce the risk of drug resistance, and optimize therapeutic effects through multiple mechanisms. The combination of guselkumab (anti-IL-23) and golimumab (anti-TNF-α) represents a promising approach for chronic inflammatory bowel diseases due to their synergistic effects on complementary inflammatory pathways. Guselkumab reduces chronic inflammation by inhibiting the IL-23/Th17 axis, while golimumab neutralizes TNF-α, targeting acute inflammation and leukocyte recruitment. Thus, this dual approach acts on both acute and chronic inflammation, potentially leading to better clinical and endoscopic remission rates and reduced recurrences. The DUET-CD trial (NCT05242471) is a phase IIb randomized, double-blind, active- and placebo-controlled, multicenter study designed to evaluate the efficacy and safety of induction and maintenance combination therapy with guselkumab and golimumab in patients with moderately to severely active CD. Participants enrolled were randomized to receive guselkumab SC, golimumab SC, JNJ-78934804 (a combination of guselkumab and golimumab), or a placebo. The primary endpoints are clinical remission (defined by CDAI score) and endoscopic response (defined by SES-CD score) at week 48. Initial results are expected in 2025 [59].

The VICTRIVA trial (NCT06227910) is a phase IIIb randomized, double-blind, placebo-controlled trial designed to evaluate the efficacy and safety of combination therapy with vedolizumab (which selectively targets integrin α4β7) and upadacitinib (a Janus kinase 1 [JAK-1] inhibitor) compared with vedolizumab monotherapy in participants with moderately to severely active CD. In the induction phase, patients will be randomized in a 1:1 ratio into two treatment arms. The first group will receive 300 mg of vedolizumab IV and 45 mg of oral upadacitinib, while the second group will receive the same dose of vedolizumab and placebo, for a total of 12 weeks. At the end of the 12-week induction period, participants who achieve a CDAI reduction of at least 70 points from baseline will be eligible to participate in the maintenance phase, during which they will receive 300 mg of IV vedolizumab every 8 weeks for an additional 40 weeks. The study endpoints include clinical remission (defined as CDAI <150) at weeks 12 and 52, and endoscopic response (defined as SES-CD reduction by ≥50% from baseline) at weeks 12 and 52. The trial has not yet started recruitment, and results are expected around 2027 [60]. From the available data, the infection and reactivation rates of latent infections are slightly higher than those available for the single agents. Patients receiving combination therapy should be closely monitored for signs of infection, with screening for tuberculosis and hepatitis B and C being particularly important. Prophylactic treatments and vaccinations should be considered before starting combination therapy.
pharmaceuticals-18-00190-t001_Table 1Table 1Studies exploring novel therapies for Crohn’s disease.ClassDrugRoute of AdministrationStudy AcronymPhaseStudy DesignResultsReferenceAnti IL-23RisankizumabIVADVANCE and MOTIVATEIIIMulticenter, randomized, double-blind, placebo-controlled induction study. Patients received risankizumab at 600 or 1200 mg or a placebo for 12 weeks.In ADVANCE study, primary endpoint occurred in 45% (in 600 mg group) and 42% (in 1200 mg group) vs. 25% in placebo group (*p* < 0.0001). In MOTIVATE study, primary endpoint occurred in 42% (in 600 mg group) and 40% (in 1200 mg group) vs. 20% in placebo group (*p* < 0.0001).[22]

SCFORTIFYIIIMulticenter, randomized, double-blind, placebo-controlled maintenance and open-label extension study. Patients received risankizumab at 180 mg or 360 mg for 52 weeks.Primary endpoint occurred in 55% (in 180 mg group; *p* = 0.048) and 52% (in 360 mg group; *p* = 0.025) vs. placebo.[23]

IVSEQUENCEIIIbMulticenter, randomized, blinded study. Patients received risankizumab or ustekinumab for 48 weeks.Risankizumab demonstrated equivalence to ustekinumab in achieving clinical remission at week 24 and superiority in achieving endoscopic remission at week 48 (*p* < 0.001).[24]
MirikizumabIVSERENITYIIMulticenter, randomized, parallel-arm, placebo-controlled study. Patients received mirikizumab at 200, 600, or 1000 mg or a placebo for 12 weeks.Primary endpoint occurred in 25.8% (in 200 mg group; *p* = 0.079), 37.5% (in 600 mg group; *p* = 0.003), and 43.8% (in 1200 mg group (*p* < 0.001) vs. placebo.[26]

IV and SCVIVID IIIIMulticenter, randomized, double-blind, placebo- and active-controlled, treat-through study. Patients received mirikizumab, ustekinumab or a placebo.Clinical remission was achieved in 45.4% (in experimental group) vs. 19.6% (in placebo group). Endoscopic response was achieved in 38% (in experimental group) vs. 9% (in placebo group) (*p* < 0.0001).[27]

IV and SCVIVID IIIIIMulticenter, open-label, long-term extension study.Results are not yet available.[28]
GuselkumabIVGALAXI IIIMulticenter, randomized, double-blind, placebo- and active-controlled, parallel group study. Patients received guselkumab at 200, 600, or 1200 mg or a placebo for 12 weeks.Primary endpoint occurred in every dose of guselkumab vs. placebo (*p* < 0.05).[29]

IV and SCGALAXI II/IIIIIIMulticenter, randomized, double-blind, placebo and active controlled, parallel group study. Patients received guselkumab at 200 mg IV →100 mg SC, 200 mg IV→200 mg SC or ustekinumab for 48 weeks.Primary endpoint occurred in 65.4% (in 200 mg IV →100 mg group) and 70.3% (in 200 mg IV→200 mg SC group) vs. 62.9% in ustekinumab group.[31]

IV and SCGRAVITIIIIMulticenter, randomized, double-blind, placebo-controlled study. Patients received guselkumab at 400 mg IV→200 mg SC q4w, 400 mg IV→100 mg SC q8w or a placebo for 48 weeks.Clinical remission and endoscopic response occurred in 56.1% and 41.3%, respectively (in guselkumab group), vs. 21% in placebo group.[32]Anti-TL1A antibodiesTulisokibartIVAPOLLO-CDIIaMulticenter, open-label study. Patients received PRA-023 at 1000 mg on day 1, and 500 mg at weeks 2, 6, and 10.Clinical remission and endoscopic response occurred in 49.1% and 26%, respectively (*p* < 0.001.[39,40]

IV and SCNCT06430801IIIDouble-blind, placebo-controlled to evaluate the efficacy and safety in patients with moderately to severely active CD.Results are not yet available.[42]
DuvakitugIVRELIEVE UCCDIIbRandomized, double-blind, dose-ranging study. Patients received low and high doses of duvakitug and a placebo in a 1:1:1 ratio for 14 weeks.Primary endpoint occurred in 26.1% (low dose of duvakitug), 47.8% (high dose), and 13% (placebo).[43]Anti miR-124ObefazimodOralENHANCE-CDIIbDouble-blind, placebo-controlled study. 12 weeks of induction followed by a 40-week maintenance period, and a 48-week extension phase.Results are not yet available.[49]

OralNCT03905109IIb/IIIMulticenter, randomized, placebo-controlled study. It consists in an induction phase followed by a maintenance phase, with a total treatment duration of 52 weeks.Results are not yet available.[50]S1P receptor modulatorsAmiselimodOralNAIIaMulticenter, randomized, double-blind, placebo-controlled. Patients were randomized to: amiselimod at 0.4 mg vs. a placebo over 12 weeksAmiselimod did not demonstrate superiority over a placebo in inducing a clinical response.[53]
OzanimodOralSTEPSTONEIIUncontrolled, multicenter trial comprising a 12-week period in which patients received an increasing dose of ozanimod.At week 12, endoscopic response, clinical remission, and clinical response were observed in 23.2%, 39.2%, and 56.5% of patients, respectively.[54]

OralYELLOWSTONEIIIA program comprising two randomized, double-blind, placebo-controlled induction studies (NCT03440372 and NCT03440385), a maintenance study (NCT03464097), and an open-label extension study (NCT03467958).Results are not yet available.[56]
EtrasimodOralCULTIVATEII/IIIA program comprising five substudies focusing on induction dosing (substudies A and 1), maintenance (substudy 3), therapeutic response (substudy 2), and long-term effects (substudy 4).Clinical remission rates of 31% and 44% with 2 mg and 3 mg doses of etrasimod, respectively.[57,58]Combination therapyGuselkumab + GolimumabSCDUET-CDIIbRandomized, double-blind, active- and placebo-controlled, multicenter study. Patients received guselkumab, golimumab, JNJ-78934804, or a placeboResults are not yet available.[59]
Vedolizumab+ UpadacitinibIVVICTRIVAIIIbRandomized, double-blind, placebo-controlled study. A first group of patients will receive 300 mg of intravenous vedolizumab and 45 mg of oral upadacitinib, while the second group will receive the same dose of vedolizumab and a placebo, for a total of 12 weeks.Results are not yet available.[60]

## 4. Ulcerative Colitis

### 4.1. Anti-Interleukin-23 (IL-23)

The phase III INSPIRE trial (NCT03398148) was a multicenter, randomized, double-blind, placebo-controlled induction study designed to evaluate the efficacy and safety of risankizumab as an induction treatment for moderately to severely active UC (Table 2). A total of 975 patients were enrolled and randomized in a 2:1 ratio to receive either placebo or 1200 mg of risankizumab every 4 weeks for 12 weeks. By week 12, clinical remission (defined as stool frequency score ≤1, no worse than baseline; rectal bleeding score of 0; and endoscopic subscore ≤1 without friability) was achieved in 20.3% of patients in the risankizumab group, compared to 6.2% in the placebo group [61]. From a safety perspective, risankizumab was well tolerated, with no serious adverse events reported. Common adverse events included anemia, exacerbation of UC, and COVID-19 infection [61].

The COMMAND maintenance study (NCT03398135) was a phase III, multicenter, randomized, double-blind, placebo-controlled trial evaluating the efficacy and safety of risankizumab as a maintenance therapy in patients who achieved clinical remission in the INSPIRE study. A total of 548 patients were randomized 1:1:1 to receive 180 mg (*n* = 179) or 360 mg (*n* = 186) of risankizumab or a placebo (*n* = 183) every 8 weeks for 52 weeks. At the end of the maintenance period, clinical remission rates were 40.2% in the 180 mg group, 37.6% in the 360 mg group, and 25.1% in the placebo group [62]. The safety results were consistent with those observed in INSPIRE, with no new adverse events reported.

The phase III LUCENT I trial (NCT03518086) was a multicenter, randomized, double-blind, parallel, placebo-controlled study evaluating the safety and efficacy of mirikizumab as an induction treatment in patients with moderately to severely active UC during 12 weeks of treatment. A total of 1162 patients were randomized in a 3:1 ratio to receive mirikizumab at 300 mg (*n* = 868) or a placebo (*n* = 294) intravenously. At week 12, clinical remission rates (defined as a modified Mayo stool frequency subscore of 0 or a stool frequency subscore of 1 with a decrease of at least 1 point from baseline, a rectal bleeding subscore of 0, and a Mayo endoscopic subscore (MES) of 0 or 1) were higher in the mirikizumab group compared to the placebo group (24.2% vs. 13.3%) [63].

The phase III LUCENT II trial (NCT03524092) was a multicenter, randomized, double-blind, parallel, placebo-controlled study evaluating the safety and efficacy of mirikizumab as a maintenance treatment in patients with moderately to severely active UC who responded to induction therapy in the LUCENT I study. A total of 544 participants were randomized in a 2:1 ratio to receive mirikizumab at 200 mg (*n* = 365) or a placebo (*n* = 179) subcutaneously for 40 weeks. At week 40, clinical remission rates were higher in the mirikizumab group compared to the placebo (49.9% vs. 25.1%) [63].

Both LUCENT I and II showed good safety, with only 1% of patients experiencing opportunistic infections such as herpes zoster [63]. Of note, mirikizumab was effective as an induction and maintenance therapy even in patients who had failed previous anti-TNF treatments. The phase III LUCENT III trial was a multicenter, open-label extension study designed to evaluate the long-term efficacy and safety of mirikizumab in patients with moderately to severely active UC who had achieved clinical remission in the LUCENT II study. Enrolled participants were treated with mirikizumab at 200 mg subcutaneously for an additional 52-week period. At week 104, clinical and endoscopic remission rates were higher in the mirikizumab group compared to the placebo [64].

Guselkumab is currently being investigated in UC. The phase IIb QUASAR trial (NCT04033445) is a multicenter, randomized, double-blind, placebo-controlled, parallel-group study evaluating the efficacy and safety of guselkumab in patients with moderately to severely active UC. Patients were randomized to receive 200 mg or 400 mg of guselkumab, or a placebo. At the end of the 12-week induction phase, clinical response rates were significantly higher in the guselkumab groups compared to the placebo (200 mg: 61.4%; 400 mg: 60.7%; placebo: 27.6%) [65].

The phase III QUASAR trial is a multicenter, randomized, double-blind, placebo-controlled induction and maintenance study. In the induction study, 701 patients were randomized in a 3:2 ratio to receive guselkumab at 200 mg intravenously (*n* = 421) or a placebo (*n* = 280) for a total of 12 weeks. At week 12, clinical remission was observed in 23% and 8% of participants treated with guselkumab and the placebo, respectively. In the maintenance study, patients who had responded during the induction study were randomized in a 1:1:1 ratio to receive guselkumab at 200 mg SC every 4 weeks, guselkumab at 100 mg SC every 8 weeks, or a placebo for a total of 44 weeks. At week 44, clinical remission was achieved in 50%, 45%, and 19% of the participants treated with guselkumab at 200 mg every 4 weeks, at 100 mg every 8 weeks, and the placebo, respectively [66].

These results demonstrate the efficacy and safety of guselkumab as a viable therapeutic option for UC [62]. Ongoing trials investigating novel therapies for UC are summarized in Table 2.

### 4.2. Anti-TL1A Antibodies

PF-06480605 is a human IgG1 monoclonal antibody directed against TL1A. Its mechanism of action is to bind to TL1A, thereby neutralizing the TL1A-DR3 interaction and subsequent signaling. One of the first trials, conducted between 2013 and 2015, was NCT01989143, a randomized, double-blind, placebo-controlled phase I study. The study aimed to evaluate the safety, tolerability, pharmacokinetics, target binding, and immunogenicity of PF-06480605 in healthy volunteers using single and multiple ascending doses. Results from the observation period showed that the drug was safe and well tolerated, with good target binding. Increasing doses of PF-06480605 proportionally increased soluble TL1A levels, effectively preventing its binding to DR3. Common side effects included headache and abdominal pain [67].

The phase IIa TUSCANY study, conducted between 2016 and 2019, was a multicenter, single-arm trial designed to evaluate the efficacy and safety (as defined by the incidence of adverse events) of PF-06480605 in subjects with moderate-to-severe ulcerative colitis. Participants (*n* = 50) received 500 mg of PF-06480605 intravenously every two weeks for a total of seven doses, followed by an additional 14-week observation period after the last dose. At the end of the study, the most common adverse events were exacerbation of disease and arthralgia. Efficacy endpoints included endoscopic improvement (defined as MES ≥ 1 without friability) in 38% of patients, endoscopic remission (MES =0) in 10%, and clinical remission (Mayo total score ≤ 2) in 24%. In addition, reductions in fecal calprotectin and CRP levels were observed. A potential limitation of the study is that 82% of patients developed anti-drug antibodies (ADAs), and 10% exhibited neutralizing antibodies. The high immunogenicity could compromise the efficacy and tolerability of the treatment [68,69].

The phase IIb TUSCANY 2 study, conducted between 2019 and 2022, was a multicenter, randomized, double-blind, placebo-controlled trial. Enrolled participants (*n* = 246) underwent an induction period with different doses of PF-06480605 (50 mg, 150 mg, and 450 mg) for 14 weeks, followed by a 56-week follow-up period. The study was designed to evaluate the efficacy, safety, and pharmacokinetics of PF-06480605 in patients with moderate-to-severe UC. At week 14, endoscopic improvement, endoscopic remission, and clinical remission were achieved in 36%, 11%, and 29% of patients, respectively. At week 56, these endpoints were achieved in 50%, 21%, and 36% of patients, respectively. The drug was well tolerated across all dose groups and demonstrated a favorable safety profile [70].

Tulisokibart is another humanized IgG1 monoclonal antibody targeting TL1A. One of the key studies investigating tulisokibart was the ARTEMIS-UC trial, a phase II, multicenter, randomized, double-blind, placebo-controlled trial designed to evaluate the efficacy and safety of tulisokibart in a 12-week induction treatment in patients with moderate-to-severe ulcerative colitis. Patients (*n* = 135) were randomized 1:1 to receive a placebo or tulisokibart intravenously (1000 mg on day 1, 500 mg at weeks 2, 6, and 10). The primary endpoints included clinical remission (defined as rectal bleeding subscore of 0 and stool frequency subscore <= 1) and endoscopic improvement (defined as endoscopic subscore ≤ 1 with no friability). At week 12, clinical remission was achieved in 26.5% of patients in the tulisokibart group (compared to 1.5% in the placebo group), while endoscopic improvement was observed in 36.8% of patients (compared to 6% in the placebo group). Additionally, reductions in symptoms such as bowel urgency, rectal bleeding, and stool frequency were reported. No serious adverse events occurred during the trial [71]. Following the 12-week induction period, participants had the option to enter into the ARTEMIS-UC extension study. Participants were classified as responders to the induction (defined as a reduction of ≥2 points and ≥30% in the modified Mayo score (MMS) from baseline, with a reduction of ≥1 in the rectal bleeding subscore or an absolute rectal bleeding subscore of ≤1 at week 12) or non-responders. Induction responders were randomized to receive tulisokibart at 250 mg (*n* = 25) or 100 mg (*n* = 22). At week 50, induction responders maintained efficacy endpoints, with greater efficacy observed in the group receiving 250 mg of tulisokibart compared to the 100 mg group. The drug continued to demonstrate a favorable safety profile [72].

Duvakitug is another monoclonal antibody targeting TL1A. The NCT05499130 trial is a phase IIb, randomized, double-blind, 14-week study designed to assess the efficacy, safety, and tolerability of TEV-48574 in adult patients with UC or CD. For UC, the primary efficacy endpoint is clinical remission (defined as an MMS ≤ 2 points), while for CD, the endpoint is endoscopic response (defined as a ≥50% reduction in the baseline SES-CD score). Initial data from this trial are expected by the end of 2024 [73].

The NCT05668013 trial is a phase IIb, randomized, double-blind extension study designed to evaluate the efficacy of two different maintenance dose regimens of duvakitug SC administered every four weeks in patients with IBD. Preliminary data are expected in 2025 [74].

SPY002 is a novel long-half-life IgG1 monoclonal antibody characterized by high selectivity and affinity for TL1A. This antibody effectively and durably blocks the TL1A–DR3 interaction, positioning it as a promising therapy for IBD. However, further preclinical and clinical studies are required to establish its safety and efficacy [75].

Although these anti-TL1A therapies are still under clinical investigation, they represent a potential breakthrough in the management of IBD.

### 4.3. Obefazimod

Over the years, phase I studies, including NCT05032560 and NCT02792686, have been conducted to investigate the safety and tolerability of ABX464 in healthy volunteers. The drug was found to be safe and well tolerated, with mild gastrointestinal disturbances and headaches being the most commonly reported adverse events [76].

The phase IIa study NCT03368118, conducted by Vermeire et al. in 2019, was a multicenter, randomized, double-blind, placebo-controlled study. Thirty-two patients were randomized 2:1 to receive a daily oral dose of 50 mg of ABX464 or a placebo for an 8-week induction period. This was followed by a long-term extension phase in which patients continued to receive the same daily dose of 50 mg. The primary endpoint of the study was safety, defined as the rate of adverse events, while secondary endpoints included clinical remission (MES ≤ 2 with no individual subscore > 1), endoscopic improvement (MES of 0 or 1), and clinical response (≥3-point reduction in MCS and ≥30% change from baseline). At week 8, clinical remission, clinical response, and endoscopic improvement were achieved in 35%, 70%, and 50% of patients treated with ABX464, respectively, compared to 11%, 33%, and 11% in the placebo group. Of the 29 patients who completed the induction phase, 22 continued into the long-term extension phase. At month 12, 55% (*n* = 12) were in clinical remission, and at month 24, 66.7% of these patients were in sustained remission. The most common adverse events were headache and abdominal pain (each reported in 17.4% of cases). Despite the promising results, the study did not achieve statistical significance in the primary efficacy endpoints [77].

The phase IIb study NCT04023396, initiated in 2020 by Vermeire et al., was a multicenter, randomized, double-blind, placebo-controlled trial aimed at evaluating the efficacy and safety of three doses (25 mg, 50 mg, and 100 mg) of ABX464, administered once-daily in patients with moderate-to-severe UC who had failed conventional therapy or biologics. Of the 254 patients enrolled, 64 were assigned to the 100 mg dose, 63 to the 50 mg dose, 63 to the 25 mg dose, and 64 to the placebo. The primary endpoint was to evaluate the change from baseline in the MMS after an 8-week treatment period. At the end of this period, a statistically significant improvement in MMS was observed in all three dose groups compared to the placebo. This result was also confirmed for the endoscopic objective, drop of fecal calprotectin levels, clinical response, and clinical remission. The effects of ABX464 treatment at week 16 were similar to those seen at week 8. In this study, the main adverse events were headache and nausea, with higher percentages in the 100 mg group, suggesting a possible dose–response relationship [78].

The phase IIb study published in January 2024, conducted by Sands et al., aimed to evaluate the efficacy of obefazimod in patients with moderately to severely active UC at weeks 8, 48, and 96, stratified by induction dose. Of the 217 patients enrolled, 58 received an induction dose of 25 mg, 51 received 50 mg, 53 received 100 mg, and the remaining 55 received a placebo. After the 8-week induction period, all patients in each group received a daily maintenance dose of 50 mg of obefazimod. The final results showed that all patients receiving the maintenance dose continued to improve at weeks 48 and 96 regardless of the induction dose [79].

Another phase IIa and IIb study, also published in January 2024 and conducted by Dulai et al., aimed to evaluate the efficacy and safety of a 25 mg maintenance dose of obefazimod in patients with UC after 8 weeks of induction treatment with a 50 mg dose. The results showed that patients who reduced their daily dose of obefazimod to 25 mg continued to achieve good disease control after one year of treatment [80].

Phase III trials are currently ongoing in the ABTECT program [NCT05507203, NCT05507216, NCT05535946]. These multicenter, randomized, double-blind, placebo-controlled studies aim to evaluate the efficacy and safety of ABX464 at doses of 25 or 50 mg once daily as an induction treatment in patients with moderately to severely active UC. Preliminary results from these studies are expected to be published in 2025 [81,82,83].

Although larger studies are needed to confirm these findings, current evidence suggests that obefazimod effectively reduces gastrointestinal inflammation, relieves associated symptoms, induces disease remission, and maintains remission, all with a favorable safety profile [45].

### 4.4. NLRX1 Agonists

In recent years, immunometabolism has gained increasing recognition for its dual role in regulating extracellular immune responses and intracellular metabolism. In this context, 1,3,5-tris(6-methylpyridin-2-yloxy) benzene (NX-13), an orally administered immune metabolite, is being investigated as an agonist of NLRX1 (NOD-like receptor X1). NLRX1 is a member of the NOD-like receptor (NLR) family, with regulatory and anti-inflammatory functions that provide protection against IBD. NLRs are a family of receptors involved in recognizing cytosolic patterns, monitoring metabolic stress, distinguishing between pathogenic bacteria and beneficial symbionts, and detecting inflammatory signals [84]. Among NLRs, NLRX1 stands out for its ability to control inflammation and interferon production in response to pathogens, while inhibiting NF-κB activity (Figure 1).

Preclinical studies in animal models showed that mice with a loss-of-function mutation in NLRX1 had significantly worse disease activity, including severe inflammation and intestinal mucosal fibrosis, than their counterparts without the mutation. This suggests that loss of NLRX1 leads to excessive inflammatory signaling mediated by chemokines and cytokines [85].

NX-13 is a low-systemic-absorption drug that specifically targets NLRX1 in the gut to reduce the risk of systemic immune-related side effects commonly seen with other autoimmune therapies. Its mechanism of action involves upregulation of NLRX1, which inhibits CD4+ T cell differentiation into Th1 subsets (reducing TNFα+ and IFNγ+ levels) and Th17 cells. It also increases oxidative phosphorylation while decreasing anaerobic glycolysis, upregulates antioxidant enzymes (e.g., Gpx1, Gstm1, Txnrd1), and decreases NF-κB and reactive oxygen species (ROS) activation [86].

Studies in murine models of UC showed that treatment with NX-13 reduced disease activity, fecal calprotectin, leukocyte infiltration, and epithelial erosion in intestinal tissue. It also increased colonic NLRX1 expression, consistent with its proposed mechanism of action [85,86].

The phase Ib study NCT04862741, conducted from April 2021 to June 2022, was a multicenter, randomized, double-blind, placebo-controlled study evaluating the safety, tolerability, and pharmacokinetics (i.e., NX-13 concentrations in colon tissue, urine, plasma, and feces) of NX-13 in patients with moderately active UC. After a 4-week screening period, patients entered a 4-week treatment period, during which they received oral NX-13 tablets once daily. Of the 36 patients enrolled, 11 received 250 mg IR (immediate release), 10 received 500 mg IR, 11 received 500 mg DR (delayed release), and 4 received a placebo. NX-13 was found to be safe and well tolerated, with no serious adverse events or deaths reported. Mild to moderate adverse events were more common in the 500 mg DR group. However, the short duration of the study limited the ability to assess the long-term safety of the drug [87].

Pharmacokinetic analyses showed that NX-13 was most concentrated in the colon and feces, especially at the 500 mg DR dose, with minimal systemic exposure (detectable levels in urine and plasma were low). Immunohistochemical analysis confirmed target engagement and showed upregulation of NLRX1 at all doses. The exploratory clinical endpoints were clinical response based on the total Mayo score (including rectal bleeding), stool frequency, MES, and physician’s global assessment (PGA). Analysis of the data showed that the strongest clinical response was observed in the group of patients receiving the 250 mg IR dose. At week 4, a clinical response was seen in 8 out of 11 patients treated with 250 mg IR, 4 out of 10 patients treated with 500 mg IR, and 3 out of 11 patients treated with 500 mg DR, while no placebo-treated patients achieved a clinical response. In line with the clinical response, reductions in fecal calprotectin and histological disease activity were also observed [87].

The phase IIa study NCT05785715 is a multicenter, randomized, double-blind, placebo-controlled trial designed to evaluate the clinical disease activity and safety of oral NX-13 in patients with moderate-to-severe UC. A total of 80 patients were enrolled and divided into three groups: placebo, NX-13 250 mg, and NX-13 750 mg. The study started in 2023, with top-line results expected in 2025 [88].

Further studies are needed to confirm the promising findings of NX-13 in preclinical and early clinical trials. This novel molecule, with its targeted action on immunometabolism, offers a promising therapeutic option for IBD. While current research is focused on UC, its localized action and low systemic absorption suggest potential efficacy in the treatment of CD.

The mechanism of action of NX-13 involves the activation of NLRX1, a mitochondrial-associated receptor, which sequesters adaptor molecules like TRAF6 and MAVS to regulate intestinal inflammation. By suppressing NF-κB signaling and promoting epithelial barrier integrity, NX-13 mitigates inflammatory pathways and modulates immune responses in the gut. Symbols: Red arrows indicate the effects of NX-13; black arrows represent activation; red X marks indicate inhibition of pathways or molecules.

### 4.5. RIPK Inhibitors

Although much research into IBD has focused on the development of drugs to inhibit immune responses, intestinal epithelial cells (IECs) have emerged as critical players in the pathogenesis of these diseases. IECs are essential for nutrient absorption, pathogen defense, and immune regulation. Dysregulated gene expression in IECs has been shown to lead to aberrant activation of inflammatory responses. A key role in this process is played by receptor-interacting protein kinases (RIPKs), a family of seven proteins (RIPK1-7) that share a homologous serine-threonine kinase domain but possess distinct functional domains. RIPKs act as regulatory molecules in several biological pathways, in the regulation of necroptosis. Their activity involves downstream signaling from receptors such as TNFR1 (TNF-alpha receptor) and NOD1/NOD2 [89].

Necroptosis is a programmed form of cell death that activates specific signaling pathways, leading to cell lysis. During necroptosis, the release of danger-associated molecular patterns (DAMPs) triggers the activation of the immune system and subsequent inflammation [90]. Recent studies in patients with severe IBD and those unresponsive to conventional biologics (e.g., anti-TNF and anti-IL-23) have highlighted excessive mucosal necroptosis mediated by RIPK1 and RIPK3 [91]. Under conditions of cellular stress, RIPK1 and RIPK3 interact to phosphorylate mixed lineage kinase domain-like protein (MLKL), which translocates to the cell membrane, disrupting its integrity and releasing DAMPs that amplify inflammation. RIPK2, another member of the RIPK family, plays a key role in activating the NF-κB and MAPK pathways, leading to the production of inflammatory cytokines such as IL-6 and TNF-α [89,92]. While these molecules are essential for defense against pathogens under normal conditions, their dysregulation and hyperactivation contribute to the pathogenesis of IBD.

A strong correlation between RIPK3 expression and CD has been demonstrated. Higher RIPK3 levels are observed in patients with severe disease (as assessed by the CDAI), upper gastrointestinal involvement, or severe forms such as stricturing and penetrating phenotypes, as well as perianal disease. Conversely, patients with minimal disease activity or isolated terminal ileum involvement have lower RIPK3 expression [90].

GSK2982772, a targeted RIPK1 inhibitor, is designed to prevent cell death and the release of inflammatory cytokines. The phase IIa clinical trial NCT02903966 was a multicenter, randomized, double-blind, placebo-controlled study with an open-label extension. The study was designed to evaluate the safety, tolerability, pharmacokinetics, pharmacodynamics, and preliminary efficacy of GSK2982772 in patients with active UC [93]. The study, conducted at 23 sites in seven countries (November 2016–June 2019), followed a two-phase treatment regimen, with each phase lasting 42 days, followed by a 28-day follow-up period. In the first phase, 36 patients were randomized 2:1 to receive either GSK2982772 (60 mg, *n* = 24) or a placebo (*n* = 12) three times daily. In the second phase, all participants received open-label GSK2982772 (60 mg) three times daily. At the end of the study, no significant differences were observed between the treatment groups, although the small sample size may have limited the ability to detect meaningful differences [93].

In addition, NCT04978493 is a phase IIa, randomized, double-blind, placebo-controlled study designed to evaluate the safety, efficacy, pharmacokinetics, and pharmacodynamics of BI 706321 administered orally for 12 weeks in patients with CD undergoing ustekinumab induction therapy. Participants were randomized to receive BI 706321 in combination with ustekinumab or a placebo plus ustekinumab. The study was completed in August 2024, but the results are still pending [94].

In summary, RIPK molecules play a fundamental role in the regulation of intestinal inflammation and cell death in IBD, making them an attractive therapeutic target for these complex and challenging diseases. Although initial studies such as those with GSK2982772 and BI 706321 provide valuable insights, further research with larger sample sizes is essential to validate these findings and optimize treatment strategies.
pharmaceuticals-18-00190-t002_Table 2Table 2Studies exploring novel therapies for ulcerative colitis.ClassDrugRoute of AdministrationStudy AcronymPhaseStudy DesignResultsReferenceAnti IL-23RisankizumabIVINSPIREIIIMulticenter, randomized, double-blind, placebo-controlled, parallel-group study. Patients received risankizumab at 1200 mg or a placebo every 4 weeks for 12 weeks.Primary endpoint occurred in 20.3% of patients in the risankizumab group, compared to 6.2% in the placebo group (*p* < 0.00001).[61]

SCCOMMANDIIIMulticenter, randomized, doubl-blind, placebo-controlled study. Patients received risankizumab at 180 or 360 mg or a placebo every 8 weeks for 52 weeks.Primary endpoint occurred in 40.2% in the 180 mg group, 37.6% in the 360 mg group, and 25.1% in the placebo group (*p* < 0.001).[62]
MirikizumabIVLUCENT IIIIMulticenter, randomized, double-blind, parallel, placebo-controlled study. Patients received mirikizumab at 300 mg or a placebo intravenously for 12 weeks.Clinical remission rates were higher in the mirikizumab group compared to the placebo group (24.2% vs. 13.3%) (*p* < 0.001).[63]

SCLUCENT IIIIIMulticenter, randomized, double-blind, parallel, placebo-controlled study. Patients received mirikizumab at 200 mg or a placebo subcutaneously for 40 weeks.Clinical remission rates were higher in the mirikizumab group compared to the placebo (49.9% vs. 25.1%) (*p* < 0.001).[63]

SCLUCENT IIIIIIMulticenter, open-label extension study. Patients received mirikizumab at 200 mg subcutaneously for 52 weeks.Clinical and endoscopic remission rates were higher in the mirikizumab group compared to the placebo.[64]
GuselkumabIVQUASARIIbMulticenter, randomized, double-blind, placebo-controlled, parallel-group study. Patients received guselkumab at 200 or 400 mg or a placebo for 12 weeks.Clinical response rates were significantly higher in the guselkumab groups compared to the placebo (200 mg: 61.4%; 400 mg: 60.7%; placebo: 27.6%).[65]

IV and SC
IIIMulticenter, randomized, double-blind, placebo-controlled induction and maintenance study. In the induction study, patients received guselkumab at 200 mg or a placebo. In the maintenance study, patients received guselkumab at 200 mg SC, 100 mg SC, or a placebo.In the induction study, clinical remission occurred in 23% and 8% of participants treated with guselkumab and the placebo, respectively. In the maintenance study, clinical remission was achieved in 50%, 45%, and 19% of participants treated with guselkumab at 200 mg every 4 weeks, 100 mg every 8 weeks, and the placebo, respectively.[66]Anti-TL1A antibodiesPF-06480605IV and SCTUSCANYIIaMulticenter, single-arm, open-label study. Patients received 500 mg IV PF-06480605 every 2 weeks, 7 doses in total, with a 14-week follow-up period.Endoscopic improvement occurred in 38% of patients; endoscopic remission occurred in 10%; clinical remission occurred in 24%.[68,69]

SCTUSCANY 2IIbMulticenter, randomized, double-blind, placebo-controlled study. Patients received PF-06480605 at 50, 150, or 450 mg for 14 weeks.Endoscopic improvement, endoscopic remission, and clinical remission were achieved in 36%, 11%, and 29% of patients, respectively (at week 14). These endpoints were achieved in 50%, 21%, and 36% of patients, respectively (at week 56).[70]
TulisokibartIVARTEMIS-UCIIRandomized, double-blind, placebo-controlled study. Patients were randomly administered IV PRA023 at 1000 mg on day 1, 500 mg at weeks 2, 6, and 10, or a placebo for 12 weeks.Clinical remission was achieved in 26.5% of patients in the tulisokibart group (compared to 1.5% in the placebo group), while endoscopic improvement was observed in 36.8% of patients (compared to 6% in the placebo group).[71]
DuvakitugSCNCT05668013IIbRandomized, double-blind extension study. Patients received two different maintenance dose regimens of subcutaneous TEV-48574 every four weeks.Results are not yet available.[73]Anti miR-124ObefazimodOralNCT03368118IIaMulticenter, randomized, double-blind, placebo-controlled study. Patients received ABX464 at 50 mg or a placebo for 8 weeks.Clinical remission, clinical response and endoscopic improvement were achieved in 35%, 70%, and 50% of patients treated with ABX464, respectively, compared to 11%, 33%, and 11% in the placebo group.[77]

OralNCT04023396IIbMulticenter, randomized, double-blind, placebo-controlled study. Patients received ABX464 at 25, 50, or 100 mg or a placebo for 8 weeks.Primary endpoint was observed in all three dose groups compared to the placebo.[78]


NCT05507203NCT05507216 NCT05535946IIIMulticenter, randomized, double-blind, placebo-controlled studies. Patients received ABX464 at 25 or 50 mg once daily.Results are not yet available.[81,82,83]NLRX1 AgonistNX-13OralNCT05785715IIaMulticenter, randomized, double-blind, placebo-controlled study. Patients received NX-13 at 250 or 750 mg or a placebo.Results are not yet available.[88]RIPK1 InhibitorGSK2982772OralNCT02903966IIaMulticenter, randomized, double-blind, placebo-controlled study with an open-label extension. In the first phase, patients received GSK2982772 at 60 mg or a placebo; in the second phase, all patients received GSK2982772 at 60 mg.No significant differences were observed between the treatment groups.[93]

### 4.6. S1P Receptor Modulators

In UC an ongoing phase II study (NCT04857112), randomized, double-blind, and placebo-controlled, is evaluating the efficacy and safety of amiselimod over 12 weeks of treatment in patients with mild-to-moderate UC. The results of this study are not yet available [95]. Based on the results of the TOUCHSTONE and TRUE NORTH studies, ozanimod was approved by the FDA in 2021 for the treatment of patients with moderate-to-severe UC. The TOUCHSTONE study was a phase II trial that enrolled 197 patients who were randomized into three groups to receive daily doses of 0.5 mg or 1 mg of ozanimod or a placebo for 32 weeks. At week 8, the 1 mg dose demonstrated a higher rate of clinical remission (57%) compared to the placebo. At week 32, ozanimod showed superiority not only in achieving clinical remission (51%) but also in secondary endpoints, including clinical response, mucosal healing, and histological remission. These results demonstrated the efficacy of the drug over the placebo. A limitation of this study lies in the significant reduction of lymphocytes below the lower limit of normal. This could increase the risk of infections [96].

The efficacy of ozanimod was further demonstrated in the phase II extension study TOUCHSTONE OLE. In this long-term study, patients were followed for four years. At the end of the study, clinical response and clinical remission were maintained in 93% and 82% of patients, respectively. Despite the extended treatment period, the drug was well tolerated [97].

The TRUE NORTH study, a phase III trial, enrolled 645 patients who were randomized to receive either 1 mg of ozanimod daily or a placebo. At week 10, 18.4% of patients achieved clinical remission compared to 6% in the placebo group. Of those who responded to induction therapy, 37% maintained clinical remission at week 52 [98]. Long-term efficacy was further supported by the TRUE NORTH OLE study, in which participants continued treatment until week 94. At the end of the study, 91.4% of patients had achieved a clinical response and 69.1% were in clinical remission. Similar improvements were seen in endoscopic and histological endpoints. Adverse events reported included elevated liver transaminases and lymphopenia, but the drug was considered safe overall [99].

The OASIS study was a phase II, randomized, double-blind, placebo-controlled, multicenter trial designed to assess the safety and efficacy of etrasimod in patients with moderate-to-severe active ulcerative colitis. A total of 156 patients were enrolled and randomized to receive either 2 mg of etrasimod daily or a placebo for 12 weeks. At week 12, patients in the etrasimod group showed significant improvements in the MMS compared to the placebo. Clinical response (51% vs. 33%), clinical remission (33% vs. 8%), endoscopic improvement (42% vs. 18%), and histological remission (20% vs. 6%) were all higher in the etrasimod group [100]. The OASIS extension study confirmed these results. Patients who continued etrasimod therapy for 52 weeks maintained clinical response (85%), clinical remission (60%), and endoscopic improvement (69%). The most commonly reported adverse events were worsening UC and anemia [101].

A post hoc analysis of OASIS evaluated the efficacy of etrasimod at week 12 in achieving endoscopic improvement and histological remission, defined as MES ≤1 (excluding friability) combined with a Geboes score < 2.0. Patients receiving 2 mg of etrasimod achieved higher EIHR rates compared to the placebo. In addition, reductions in fecal calprotectin and CRP levels correlated with clinical and endoscopic response, highlighting these biomarkers as potential indicators of therapeutic success [102].

The phase III ELEVATE UC 12 study was a randomized, double-blind, and placebo-controlled trial evaluating the efficacy and safety of etrasimod in patients with moderate-to-severe UC who had failed prior therapy. Patients were randomized (2:1) to receive etrasimod 2 mg (*n* = 238) or a placebo (*n* = 116) orally once daily for a 12-week induction period. At the end of the study, 25% of participants in the etrasimod group achieved clinical remission (defined as a combined stool frequency score = 0 [or stool frequency = 1 with a 1-point reduction from baseline], rectal bleeding subscore = 0, and MES ≤ 1) compared to 15% in the placebo group [103]. In the phase III ELEVATE UC 52 study, 289 patients were randomized to receive etrasimod at 2 mg or a placebo for a 12-week induction period followed by a 40-week maintenance period. At week 52, clinical remission was achieved in 32% of the etrasimod group compared to 7% of the placebo group. Both ELEVATE UC 12 and ELEVATE UC 52 demonstrated reductions in fecal calprotectin and CRP levels in responders. No serious adverse events were reported, confirming the safety and tolerability of the drug [103].

The ongoing phase III ELEVATE UC OLE study aims to evaluate the long-term efficacy and safety of etrasimod in patients with moderate-to-severe UC who participated in the ELEVATE UC 12 and 52 studies. Top-line results are expected in 2029 [104]. A post hoc analysis of the ELEVATE UC 12 and 52 studies demonstrated the efficacy of 2 mg of etrasimod even in patients with isolated proctitis (defined as rectal involvement <10 cm). The results underline the versatility of etrasimod in different UC subtypes [105].

The phase III ELEVATE UC 40 JAPAN multicenter, double-blind, placebo-controlled study evaluated the efficacy and safety of etrasimod in Japanese patients with moderate-to-severe active UC. At week 12, clinical remission rates were higher in the etrasimod group (14.3%) compared to the placebo (7.1%). At week 52, these rates increased to 25.0% and 7.1%, respectively, further supporting the efficacy of etrasimod in different populations [106].

VTX002 is an oral selective S1P1 modulator being investigated for the treatment of moderate-to-severe UC. The phase II study NCT05156125 is a multicenter, randomized, double-blind, placebo-controlled trial designed to evaluate the efficacy and safety of VTX002 in patients with moderate-to-severe UC who have had an inadequate response to other biologic therapies. Patients were randomized into three groups to receive 30 mg (*n* = 73), 60 mg (*n* = 70), or a placebo (*n* = 70) once daily for 13 weeks. At the end of the treatment period, clinical remission (defined as stool frequency subscore ≤ 1, rectal bleeding subscore = 0, and MES ≤1) was achieved in 27.9% of patients receiving 60 mg of VTX002 compared to 11.4% in the placebo group. Secondary endpoints including endoscopic, clinical, and histological response rates also favored VTX002 over the placebo. In addition, a pharmacodynamic reduction in absolute lymphocyte count was observed in patients treated with VTX002, supporting its mechanism of action as an S1P1 receptor modulator. Although no serious adverse events were reported, this reduction could increase the risk of long-term infections [107].

## 5. Discussion

Several innovative small molecules and biologics have entered phase II and III clinical trials in patients with active moderate-to-severe IBD. Positive results have been reported for some of these compounds, highlighting their potential for future incorporation into standard of care.

Specifically, S1P receptor modulators act through a unique mechanism that traps lymphocytes within lymphoid tissues, which may make them particularly effective in treating UC [50]. Selective IL-23p19 monoclonal antibodies, including risankizumab, mirikizumab, and guselkumab, have demonstrated excellent safety profiles and efficacy rates even in biologic-experienced patients [20,45]. The future of drug development in IBD lies in addressing the unmet needs of patients who do not achieve remission or experience adverse effects with current therapies. Biologics such as anti-TNF agents (e.g., infliximab and adalimumab) have revolutionized treatment, but up to 30–40% of patients experience primary non-response and approximately 50% lose response over time, requiring dose escalation or switching therapies. Newer classes, such as anti-IL23 and S1P receptor modulators, have demonstrated efficacy, with clinical remission rates of 17–40% in randomized trials. Despite their promising efficacy, these drugs are not without significant risks, such as a potential increase in infections. Biologic therapies, by targeting specific immune pathways, can lead to a reduction in immune defenses [22,27]. Among the most important alerts that have emerged from studies are moderate–severe zoster, which is currently preventable by available recombinant vaccination. Overall, the rates of serious infections found in the most recent studies are superimposable or even lower than those observed for anti-TNFs.

Another concern is long-term immunosuppression, which, with therapies like TNF-α or JAK inhibitors, could theoretically increase the risk of malignancy. However, current data from clinical trials and real-world experience have disproved any significant increase in malignancy rates among patients treated with IL-23 inhibitors, such as guselkumab and mirikizumab, during the studied follow-up periods.

To minimize these risks, targeted strategies can be employed: pre-treatment assessments to evaluate infection risk (e.g., screening for tuberculosis and testing for hepatitis B/C), continuous monitoring through pharmacovigilance during therapy, and educating patients to recognize and manage early signs of adverse events effectively.

In general, the phase II and III clinical trials discussed in this review present some common limitations that warrant attention. Firstly, the observation period is often insufficient to evaluate the long-term effects of the various pharmacological classes, both in terms of efficacy and safety. Furthermore, most of these studies apply highly stringent inclusion and exclusion criteria, focusing primarily on specific disease phenotypes (moderate-to-severe forms), while overlooking milder or atypical manifestations. Another significant limitation is the small sample size, which can compromise the robustness of the results and the generalizability of the conclusions. In light of this, it is essential to increase the number of participants and extend the follow-up period to more thoroughly assess the long-term safety and efficacy endpoints.

The heterogeneity of IBD requires a shift towards personalized medicine. Advances in genomic, transcriptomic, and microbiome research are paving the way for biomarkers to guide treatment decisions, predict response, and monitor disease activity. However, translating these discoveries into clinical practice remains a challenge due to the complexity of integrating multi-omics data and ensuring cost-effectiveness.

Collaboration between academia, industry, and regulators is essential to overcome barriers to drug development, including high costs, lengthy approval processes, and safety concerns. Finally, patient-centered approaches must be prioritized. This includes addressing quality-of-life issues such as fatigue and mental health, which are often overlooked in traditional endpoints such as mucosal healing. As we move forward, promoting global collaboration and equitable access to new therapies will be essential to ensure that advances in IBD management benefit diverse patient populations. Overall, the future of IBD drug development lies in a multifaceted approach that combines innovative therapeutics, precision medicine, and holistic care to improve long-term outcomes.

## 6. Conclusions

The exact disease phenotype that is most likely to respond to each advanced therapy remains to be defined. Factors influencing the choice of therapy include concomitant rheumatological and/or dermatological conditions (e.g., selective p19 anti-IL-23), and failure of previous mechanisms (e.g., S1P modulators, obefazimod). In the future, it seems reasonable that some of these new molecules will find a place in the therapy of steroid-dependent patients (also in the first line) or as an alternative to azathioprine. More data are needed to define the use of these new molecules, in all cases “forgotten” by clinical trials, such as recent oncological history, frail or elderly patients, as well as patients with multiple surgeries or isolated proctitis, isolated small bowel diseases, and/or perianal disease.

## Figures and Tables

**Figure 1 pharmaceuticals-18-00190-f001:**
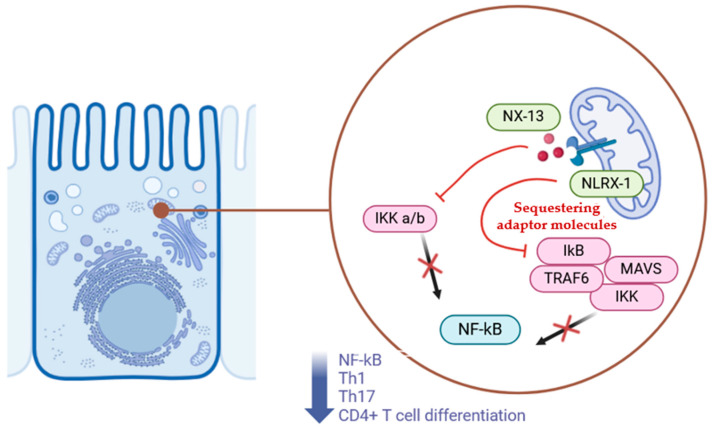
Mechanism of action of NX-13.

## Data Availability

No new data were created or analyzed in this study. Data sharing is not applicable to this article.

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
