# Peer review of "Drug Development in Inflammatory Bowel Diseases: What Is Next?"

_pharmaceuticals, 2025, doi:10.3390/ph18020190_

Round 1

Reviewer 1 Report

Comments and Suggestions for Authors

The present review addresses a significant topic in the field of inflammatory bowel diseases. In general, the manuscript is a good, valuable work and will be an interesting source for the scinetific community. However, the manuscript should undergo some revision. I would recommend the following corrections:
1. Improve the introduction, include updated epidemiological data to focus on the prevalence of inflammatory bowel disease.
2. Explain if there any limitations of current clinical trials.

3. Discuss potential barriers to implement the discussed therapies in clinical practice and compare them to existing treatment of Crohn’s disease and ulcerative colitis.
4. Explain better in the text significance of miR-124 for Crohn’s disease or other inflammatory diseases.

5. Do you have any preclinical data that suggest obefazimod’s safety and efficacy profile?
6. Critically analyze the failure of amiselimod in comparison to the success of ozanimod and etrasimod.
7. Explain why the combination of guselkumab and golimumab might be synergistic.

8. Address potential safety concerns associated with combination therapy, such as immunosuppression and infection risk.
9. Provide more detailed discussion of specific risks, such as infection rates, malignancy, and strategies.
10. Minor typographical errors can be found in the text.

Author Response

Response to the editor(s) of ‘Pharmaceuticals’

Dear Editor(s),

We sincerely thank you for giving us the opportunity to consider for re-submission and potential publication a revised version of our original manuscript entitled “Drug development in inflammatory bowel disease: what’s next?" by Petronio et al.

We kindly thank the Reviewers for the precious comments. We are pleased to know that you appreciated the topic of our manuscript. The manuscript has been significantly revised and improved according to the received suggestions. Included below you can find a point-by-point response to the remarks.

Sincerely,

Alessandro Armuzzi, Professor

alessandro.armuzzi@hunimed.eu

Reviewer 1

The present review addresses a significant topic in the field of inflammatory bowel diseases. In general, the manuscript is a good, valuable work and will be an interesting source for the scinetific community. However, the manuscript should undergo some revision. I would recommend the following corrections:
1. Improve the introduction, include updated epidemiological data to focus on the prevalence of inflammatory bowel disease.

Re: Thank you for Your comment. We added a section focusing on the prevalence of IBD in Western and newly industrialized countries.

  1. Explain if there any limitations of current clinical trials.

Re: Thank you for Your comment. We added the significant limitations of the trials discussed in our review.

  1. Discuss potential barriers to implement the discussed therapies in clinical practice and compare them to existing treatment of Crohn’s disease and ulcerative colitis.
    Re: Thank you for Your comment. We have already addressed the potential barriers and strategies to overcome them in the "Discussion" section.

  1. Explain better in the text significance of miR-124 for Crohn’s disease or other inflammatory diseases.
    Re: Thank you for Your comment. We included more details about the role of miR-124 in the text as suggested.

  1. Do you have any preclinical data that suggest obefazimod’s safety and efficacy profile?

Re: Thank you for Your comment. We added a section on preclinical data in the paragraph as suggested.

  1. Critically analyze the failure of amiselimod in comparison to the success of ozanimod and etrasimod.

Re: Thank you for Your comment. We included the probable reason for the lack of efficacy of amiselimod in the text accordinlgy.

  1. Explain why the combination of guselkumab and golimumab might be synergistic.

Re: Thank you for Your comment. We added a dedicated section in “combination therapy” paragraph.

  1. Address potential safety concerns associated with combination therapy, such as immunosuppression and infection risk.
    Re: Thank you for Your comment. We added a dedicated section in “combination therapy” paragraph.

  1. Provide more detailed discussion of specific risks, such as infection rates, malignancy, and strategies.
    Re: Thank you for Your comment. We added a dedicate section in “Discussion” paragraph.

  1. Minor typographical errors can be found in the text.

Re: Thank you for Your comment. We corrected accordingly, as suggested.

Reviewer 2 Report

Comments and Suggestions for Authors

Dear Author,

The manuscript “Drug development in inflammatory bowel diseases: what’s next?” has been thoroughly evaluated. Although the review studies have limitations as a general perspective, the manuscript offers novel insights about the future of inflammatory bowel diseases therapeutics. As is common in all studies, several minor errors have been observed in this work. It is believed that addressing the minor issues outlined below will significantly enhance the suitability of the study for publication.

1. Please continue to use your abbreviations consistently. (For example, line 34: inflammatory bowel disease (IBD) repeated in line 75-76: pathogenesis of inflammatory bowel 75 diseases (IBD).) Please check all the text for this kind of problems.

2. Check all the manuscript for the typos or grammatical errors (line 126: …ustekinumab at week 48 (56.9% vs. 31.0% [24].).

3. The paragraphs should be separated to provide consistency and possible misunderstanding. (line 161-170 and line 171-181) (an appropriate intro part could be beneficial at the beginning of the line 171.).

4. The table 1 and table 2 need to have an extra column for the references of related works. As a suggestion the route of administration could be added to enrich the content of tables.

5. The references section should be carefully checked. Before that authors for instructions should be carefully investigated. The journal names should be abbreviated according to the rules in the “authors for instructions” dossier. (For example: REF5-8, 10…etc.).

Respectfully yours.

Author Response

The manuscript “Drug development in inflammatory bowel diseases: what’s next?” has been thoroughly evaluated. Although the review studies have limitations as a general perspective, the manuscript offers novel insights about the future of inflammatory bowel diseases therapeutics. As is common in all studies, several minor errors have been observed in this work. It is believed that addressing the minor issues outlined below will significantly enhance the suitability of the study for publication.

  1. Please continue to use your abbreviations consistently. (For example, line 34: inflammatory bowel disease (IBD) repeated in line 75-76: pathogenesis of inflammatory bowel 75 diseases (IBD).) Please check all the text for this kind of problems.

Re: Thank you for Your comment. We corrected accordingly, as suggested.

  1. Check all the manuscript for the typos or grammatical errors (line 126: …ustekinumab at week 48 (56.9% vs. 31.0% [24].).

Re: Thank you for Your comment. We corrected it accordingly, as suggested.

  1. The paragraphs should be separated to provide consistency and possible misunderstanding. (line 161-170 and line 171-181) (an appropriate intro part could be beneficial at the beginning of the line 171.).

Re: Thank you for Your comment. We have increased the spacing between the various paragraphs regarding the different types of medications to avoid possible misunderstandings.

  1. The table 1 and table 2 need to have an extra column for the references of related works. As a suggestion the route of administration could be added to enrich the content of tables.

Re: Thank you for your comment. As suggested, we added a column for references and routes of administration.

  1. The references section should be carefully checked. Before that authors for instructions should be carefully investigated. The journal names should be abbreviated according to the rules in the “authors for instructions” dossier. (For example: REF5-8, 10…etc.).

Re: Thank you for Your comment. We corrected it accordingly, as suggested.